# Topoisomerase II as a Novel Antiviral Target against Panarenaviral Diseases

**DOI:** 10.3390/v15010105

**Published:** 2022-12-30

**Authors:** Tosin Oladipo Afowowe, Yasuteru Sakurai, Shuzo Urata, Vahid Rajabali Zadeh, Jiro Yasuda

**Affiliations:** 1Department of Emerging Infectious Diseases, Institute of Tropical Medicine (NEKKEN), Nagasaki University, Nagasaki 852-8523, Japan; 2Program for Nurturing Global Leaders in Tropical and Emerging Communicable Diseases, Graduate School of Biomedical Sciences, Nagasaki University, Nagasaki 852-8523, Japan; 3National Research Center for the Control and Prevention of Infectious Diseases (CCPID), Nagasaki University, Nagasaki 852-8523, Japan

**Keywords:** arenaviruses, minigenome, drug screening, topoisomerase II, panarenaviral replication inhibitors

## Abstract

Although many arenaviruses cause severe diseases with high fatality rates each year, treatment options are limited to off-label use of ribavirin, and a Food and Drug Administration (FDA)-approved vaccine is not available. To identify novel therapeutic candidates against arenaviral diseases, an RNA polymerase I-driven minigenome (MG) expression system for Lassa virus (LASV) was developed and optimized for high-throughput screening (HTS). Using this system, we screened 2595 FDA-approved compounds for inhibitors of LASV genome replication and identified multiple compounds including pixantrone maleate, a topoisomerase II inhibitor, as hits. Other tested topoisomerase II inhibitors also suppressed LASV MG activity. These topoisomerase II inhibitors also inhibited Junin virus (JUNV) MG activity and effectively limited infection by the JUNV Candid #1 strain, and siRNA knockdown of both topoisomerases (IIα and IIβ) restricted JUNV replication. These results suggest that topoisomerases II regulate arenavirus replication and can serve as molecular targets for panarenaviral replication inhibitors.

## 1. Introduction

*Arenaviridae* is a family of enveloped, negative-sense, RNA viruses [1] and is made up of four genera: *Reptarenavirus*, *Hartmanivirus*, *Antenavirus*, and *Mammarenavirus*. The *Mammarenavirus* is divided into two: Old World and New World groups [2]. The Old World group contains Lassa virus (LASV) and lymphocytic choriomeningitis virus (LCMV) [3]. New World group includes Junin virus (JUNV) and Machupo virus (MACV). Many arenaviruses are highly pathogenic to humans, causing severe hemorrhagic fever, and are classified as BSL-4 agents [4]. Arenavirus genome is composed of two segments: a small (S) segment and large (L) segment with average lengths of 3.4 kb and 7.2 kb, respectively. Each segment encodes two proteins that are separated by an intergenic region (IGR) in an ambisense orientation. The S segment encodes nucleoprotein (NP) and glycoprotein precursor complex (GPC). The L segment encodes the matrix protein (Z) and the RNA-dependent RNA polymerase L protein (L) [5].

Many *Mammarenaviruses* are considered as significant public health threats. LASV causes Lassa fever (LF), which is endemic to some West African countries, with approximately 300,000 cases and 5000 deaths annually [6]. Moreover, the mortality rate recorded in hospitals is usually 15–30% [7]. Furthermore, JUNV is the etiological agent of Argentine hemorrhagic fever (AHF), a severe zoonotic disease that is endemic to the Pampas region of Argentina. Besides, an annual incidence of 100–1000 AHF cases, with a case fatality rate (CFR) of approximately 15% when there is no treatment, has been reported [8].

Overall, arenavirus infections are considered neglected viral diseases as there are no vaccines or specific anti-arenaviral agents, with the only limited available therapeutic option for decades being ribavirin [9]. Treatment options for JUNV are restricted to the use of immune convalescent plasma with defined doses of JUNV-neutralizing antibodies [8]. Although combined off-label use of ribavirin and favipiravir is an option [10], ribavirin loses effectiveness in advanced infection and is also associated with serious side effects [11,12]. Plasma transfusion is also ineffective in advanced cases of JUNV infection, and 10% of treated patients experience late neurological complications [8,13]. Moreover, the live-attenuated JUNV vaccine Candid #1 is only allowed for use in endemic areas [14]. Thus, there is a need for identification of more effective anti-arenaviral agents, which is the objective of our study.

In this study, we screened a library of Food and Drug Administration (FDA)-approved drugs for inhibitors of arenavirus genome replication via a high-throughput screening (HTS) system using an LASV minigenome (MG). The MG system is an important tool for screening and testing antivirals against highly pathogenic viruses in the absence of high-containment laboratories [15]. Following two rounds of screening, a series of compounds were found to inhibit LASV MG activity with minimal cytotoxicity.

Among the hit compounds, pixantrone maleate, a known topoisomerase II inhibitor, had the highest selectivity index (SI) value. Multiple topoisomerase II inhibitors, including pixantrone maleate, also limited activities of LASV MG and JUNV MG as well as replication of the live-attenuated JUNV vaccine strain. Additionally, siRNA knockdown of topoisomerase II confirmed its importance for efficient replication of JUNV. These results suggest that topoisomerase II may constitute a molecular target for host-oriented panarenavirus therapeutic agents.

## 2. Materials and Methods

### 2.1. Cells

The human hepatocellular carcinoma (Huh-7), African green monkey kidney (Vero 76), adenocarcinoma human alveolar basal epithelial (A549), and baby hamster kidney (BHK-21) cell lines were obtained from the Health Science Research Resources Bank (JCRB0403, JCRB9007, JCRB0076, and JCRB9020). All cell lines were maintained in Dulbecco’s modified Eagle’s medium (DMEM, 04-2976, Fuji Film Wako, Osaka, Japan) supplemented with 10% fetal bovine serum (FBS) and 1% penicillin/streptomycin.

### 2.2. Virus and Viral Titration

The live-attenuated Candid #1 strain of JUNV was kindly provided by Dr. Juan C. de la Torre (The Scripps Research Institute, San Diego, CA, USA). The working stock was propagated and quantified via plaque assay as previously described [16].

The rescue of the recombinant LCMV (GenBank accession no. AY847350 and AY847351), rLCMV.Arm, was previously described [17].

### 2.3. Compounds

An FDA-approved drug library, (96-well)-L1300-Z382747-100 µL, containing 2595 compounds was obtained from Selleck Chemicals. Compounds were stored in water or DMSO at 1 mM stock solution at −30 °C until use. The topoisomerase II inhibitors pixantrone maleate (S5059), amonafide (S1367), idarubicin hydrochloride (S1228), ellipticine hydrochloride (S6790), and voreloxin (SNS 595) hydrochloride (S7518) were purchased from Selleck Chemicals as were proflavine (S5776), quinacrine dihydrochloride (S4255) and quinacrine dihydrochloride dihydrate (QDD) (S5435). Ribavirin (R9644) and mycophenolic acid (MPA, M5255) were obtained from Sigma-Aldrich. All compounds were dissolved in DMSO, except for quinacrine dihydrochloride, which was initially dissolved in water before further dilution in DMSO. Working solutions were stored at −30 °C; the original stocks were preserved at −80 °C.

### 2.4. Antibodies

An anti-topoisomerase II alpha + topoisomerase II beta antibody [EPR5377] (ab239984) was purchased from Abcam. An anti-rabbit IgG HRP conjugate antibody (W4011) was obtained from Promega (Madison, WI, USA), and anti-β-actin (A1978) and anti-mouse IgG (A2304) antibodies were purchased from Sigma-Aldrich.

In vivo MAb anti-LCMV nucleoprotein VL-4 (BE0106) was obtained from BioXCell (New Hampshire, USA) while a goat anti-rat IgG (H+L) cross-adsorbed secondary antibody, Alexa Fluour™ 488 (A-11006) was purchased from Invitrogen (Waltham, MA, USA).

### 2.5. LASV, JUNV, and LCMV Minigenome (MG) Construction

An S-segment LASV (GenBank accession no. MK107964) MG was designed with SnapGene software (from Insightful Science; available at snapgene.com) and synthesized by GENEWIZ, Inc. (South Plainfield, NJ, USA). The sequence contains the 3′ untranslated region (UTR), 5′UTR, and IGR in an antisense orientation (Figure 1a). The LASV GPC and NP coding sequences were deleted. A Nanoluciferase (Nluc) reporter gene, which enabled quantification of MG activity, was inserted at the NP locus. The efficiency of the MG was promoted by the addition of a G residue upstream of the 3′UTR. BbsI flanking overhangs facilitated cloning of the synthesized construct into the pHH21 vector [18], with the MG under control of the human polymerase-I promoter and a murine polymerase-I terminator. More details about the LASV MG construction are available upon request. The supporting plasmids, pCAGGS-NP and pCAGGS-L, were constructed by cloning LASV NP and LASV L into pCAGGS.

The construction of the JUNV MG and its supporting plasmids has been described previously [19]. In summary, the MG was based on the S segment of the JUNV Candid #1-S segment (GenBank accession no. AY746353) with the 3′UTR, 5′UTR, and IGR in an antisense orientation. MG activity was monitored with the Nluc reporter gene inserted into the NP locus. The MG system was completed with plasmids expressing JUNV NP (pC-Candid-NP) and JUNV L (pC-Candid-L), which were kindly provided by Dr. Juan C. de la Torre (The Scripps Research Institute) [20].

The LCMV MG was constructed based on the S segment of the Armstrong 53b strain. However, it was driven by a murine polymerase I promoter and tagged with a red fluorescent protein (RFP) instead of Nluc. The supporting plasmids, pC-LCMV-NP and pV-LCMV-L, were constructed by cloning LCMV NP and LCMV L into pCAGGS.

### 2.6. LASV, JUNV, and LCMV MG Assays

The quality of the established LASV MG assay was determined by calculating its Z’-factor, as described by Zhang et al. [21]. To determine the main signal of the LASV MG, plasmids encoding LASV NP, LASV L, and LASV MG (at a ratio of 1:2:1.67) were used to reverse-transfect Huh-7 cells, seeded at a concentration of 2 × 10^4^ cells/100 µL in a 96-well plate. The background signal was determined as described above, without the inclusion of LASV L. MG activity was measured at 48 h after incubation via Nluc expression with a Nano-Glo^®^ Luciferase Assay System (N1110, Promega) according to the manufacturer’s instructions.

The Z-factor of the LASV MG assay was evaluated to determine its suitability for HTS [21]. Cells were seeded and transfected as described above. At 12 h post-transfection, cells were treated with either DMSO or 100 µM ribavirin and MG activity was measured at 36 h post-treatment.

The LASV MG assay was validated with two-fold serial dilutions of ribavirin (starting from 200 µM) or MPA (starting from 10 µM) in triplicate. For compound screening with the LASV MG assay, treatment was with the negative control (DMSO), positive controls (100 µM ribavirin and 5 µM MPA), or library compounds (5 µM) in duplicate. For dose response assays with either LASV or JUNV MG assay, two-fold serial dilutions of compounds were introduced in triplicate. Cell seeding, transfection, compound treatment, and assessment of MG activity were performed at the same time points as described above. Cell viability was also measured at 36 h post-treatment with a CellTiter-Glo™ Luminescent Cell Viability Assay Kit (G7570, Promega) according to the manufacturer’s instructions. All values for MG activity and cell viability were normalized to those of DMSO-treated wells.

The LCMV MG assay was conducted to further assess the anti-panarenaviral property of identified compounds. BHK-21 cells were seeded at a concentration of 2 × 10^4^ cells/100 µL in a 96-well plate and reverse-transfected with plasmids encoding LCMV NP, LCMV L, and LCMV MG (at a ratio of 1:2:1.67). At 12 h post-transfection, cells were treated with compounds at the indicated concentrations in octuplicate. Fluorescence was measured at 36 h post-treatment with a Cytation 5 cell imaging multimode reader.

### 2.7. Dose Response Assay with JUNV (Candid #1 Strain)

A549 cells seeded at a concentration of 3.0 × 10^5^ cells/mL in 24-well plates for 2 days were infected with JUNV at an MOI of 0.01 for 1 h. The virus solution was then replaced with two-fold serial dilutions of compounds in triplicate. Untreated cells served as controls. Two days later, the supernatant was collected for viral RNA extraction using QIAamp Viral RNA Mini Kit (52906, QIAGEN) according to the manufacturer’s instructions. Cell viability was assessed at 48 h post-treatment.

### 2.8. Dose Response Assay with LCMV (rLCMV.Arm)

A549 cells were seeded at a concentration of 2.5 × 10^5^ cells/mL in 96-well plates for 24 *h*. Cells were then infected with LCMV at an MOI of 0,1 for 1 h. The virus solution was then replaced with two-fold serial dilutions of compounds in triplicate. Untreated wells served as controls. Then, after 24 h, media was removed and cells were fixed with 4% paraformaldehyde phosphate-buffered solution (4% PFA, 163-20145, Fuji Film Wako) for 1 h. Cells were blocked with blocking buffer (10% FBS in dilution buffer) for one hour; dilution buffer is a mixture of 15 g bovine serum albumin (BSA) and 1.5 mL triton-X100 in 500 mL of PBS(-). Cells were then stained with anti-LCMV nucleoprotein VL4 (primary antibody) and goat anti-rat IgG (secondary antibody). Fluorescence was measured with a Cytation 5 machine and collected data were analyzed using CellProfiler software.

### 2.9. siRNA Knockdown

A549 cells were reverse-transfected with topoisomerase IIα or IIβ siRNA using Lipofectamine™ RNAiMAX (13887075, Invitrogen) according to the manufacturer’s instructions. In brief, the cells were seeded in 24-well plates at a concentration of 1.25 × 10^5^ cells/mL and transfected simultaneously with a final siRNA concentration of 20 nM.

To evaluate the efficiency of knockdown, intracellular RNA was collected with Purelink™ RNA Mini Kit (12183025) three days after simultaneous knockdown and cell seeding. Topoisomerase II RNA levels was assessed with qPCR. Cell lysates were collected for Western blotting five days after siRNA knockdown.

To assess the impact of topoisomerase II knockdown on JUNV replication, cells were infected on the third day after seeding at an MOI of 0.01. After 1 h post-infection (h.p.i), the virus was washed out and replaced with DMEM supplemented with only 10% FBS (based on recommendations from Invitrogen™, manufacturers of Lipofectamine™ RNAiMAX transfection agent). The viral supernatant was collected at 48 h.p.i, and viral RNA was extracted and quantified by qPCR. Intracellular RNA was also collected on the same day and present viral RNA levels were quantified by qPCR.

To evaluate the influence of topoisomerase II knockdown on cell viability and topoisomerase II expression, cells were seeded and transfected as described above on separate plates. The medium was replaced on the fourth day of the experiment. Cell viability was measured at 48 h after replacing the medium. Cell lysis and collection for Western blot analysis was also performed.

The siRNAs were obtained from QIAGEN. The topoisomerase IIα siRNAs were TOPO2A siRNA 1 (SI03081281, target sequence: CCGCGTGGTCAAAGAGTCATT) and TOPO2A siRNA 2 (SI04384072, target sequence: AACCAGCGTGTTGAGCCTGAA); the topoisomerase IIβ siRNAs were TOPO2B siRNA 1 (SI02780736, target sequence: TCGGGCTAGGAAAGAAGTAAA) and TOPO2B siRNA 2 (SI04437377, target sequence: ATGGGCTTGTAAACTACCCAA); and control siRNA (1027280, proprietary target sequence).

### 2.10. Western Blotting

siRNA-transfected cells were lysed with lysis buffer (50 mM Tris-HCl, 62.5 mM EDTA, 1% NP-40, 0.4% deoxycholate). The lysates were resolved by 7.5% sodium dodecyl sulfate-polyacrylamide gel electrophoresis (SDS-PAGE). Western blotting using antibodies against topoisomerase IIα, topoisomerase IIβ, and β-actin was performed. The protein bands were detected with ECL Prime (GE Healthcare) and visualized with LAS3000 (GE Healthcare).

### 2.11. Quantitative Real Time-Polymerase Chain Reaction (qPCR)

RNA content in viral supernatants was quantified by qPCR using the standard curve method. Synthesis of standard RNA has been previously described [19]. qPCR was performed using One-Step TB Green PrimeScript PLUS RT-PCR Kit (RR096A, Takara Bio, Shiga, JAPAN) with primers targeting a region within the JUNV GPC (Forward: CCAACCTTTTTGCAGGAGGC, Reverse: TTCCTGCAAGCGCTAGGAAT) and an ABI 7500 Real-Time PCR system (Applied Biosystems, Massachusetts, USA).

Intracellular RNA levels of topoisomerase IIα (Forward: GTGGCAAGGATTCTGCTAGTCC, Reverse: ACCATTCAGGCTCAACACGCTG), topoisomerase IIβ (Forward: GGTCAGTTTGGAACTCGGCTTC, Reverse: AGGAGGTTGTCATCCACAGCAG), JUNV GPC (Forward: CCAACCTTTTTGCAGGAGGC, Reverse: TTCCTGCAAGCGCTAGGAAT), LASV NP (Forward: TGAGCAGAGGAAGGCGTTGT, Reverse: CCCGTCTCTTCCAGGTTGGG), LASV L (Forward: GCAGGGGAGCACTATGGGAG, Reverse: TGCTCCGATTAGGAGGCGTG), and LASV MG (Forward: CGCCAGTCCCCAACGAAATC, Reverse: CCGGGGATCCTAGGCATTTAGG) were measured by qPCR relative to that of glyceraldehyde 3-phosphate dehydrogenase (GAPDH) which served as an endogenous control. The primer sequences of GAPDH are Forward: CAAATTCCATGGCACCGTCA and Reverse: TAGTTGCCTCCCCAAAGCAC.

The intracellular RNA levels of LASV MG (+) and (-) strands were assessed with two-step RT-qPCR. Synthesis of the LASV MG (+) strand (Primer: AGACCAAGGGAGACGATGCC) and LASV MG (-) strand (Primer: CCGGGGATCCTAGGCATTTAGG) was achieved with Superscript™ IV First-Strand Synthesis System (18091050, Invitrogen) according to the manufacturer’s instructions. The (+) (Forward: GACCAAGGGAGACGATGCC, Reverse: GCTGTTCCGAGTAACCATCAACG) and (-) (Forward: CGCCAGTCCCCAACGAAATC, Reverse: CCGGGGATCCTAGGCATTTAGG) strands were then amplified using the One-Step TB Green PrimeScript PLUS RT-PCR Kit (RR096A, Takara Bio).

### 2.12. Statistical Analysis

Data collection was mostly performed with Microsoft Excel 2016. The dose response of all screened compounds was analyzed by nonlinear regression analysis. Differences among the experimental siRNA groups, differences in response of LCMV MG to compound treatment, and differences in intracellular levels of LASV MG system components were all analyzed with one-way ANOVA. The statistical significance level (α) was set to <0.05 for all experiments. All analyses were performed using GraphPad Prism version 9.3.1 for Windows (GraphPad Software, San Diego, CA, USA). Diagrams were created with the online software BioRender (BioRender.com, Date of accession: 14 October 2022).

Data obtained from Cytation 5 cell imaging multimode reader were analyzed with CellProfiler software.

## 3. Results

### 3.1. Construction, Optimization, and Validation of an LASV MG for Compound Screening

To identify arenavirus genome replication inhibitors, a LASV MG based on the LASV S segment genome was constructed (Figure 1a). A Z’-factor of 0.69 confirmed the LASV MG assay to be of good quality (Figure 1b) [21]. Next, the suitability of the assay for HTS assay was confirmed using ribavirin, a known arenavirus replication inhibitor, with a recorded Z-factor of 0.74 (Figure 1c). Thereafter, we validated the assay by assessing the dose response to ribavirin and MPA (Figure 1d). MPA is a broad-spectrum inhibitor of purine biosynthesis with activity against LASV and other arenaviruses [22]. The assay responded to ribavirin and MPA in a dose-dependent manner, with IC_50_ values of 15.42 µM and 0.37 µM, respectively.

### 3.2. Screening of FDA-Approved Compounds for Inhibitors of LASV MG Activity

After conducting the first round of screening as illustrated in Figure 2a, we identified 39 hit compounds which inhibited MG activity by at least 90% (red rectangle in Figure 2b). These compounds were rescreened, and their cytotoxicity was evaluated simultaneously. Twenty-eight of the hit compounds still exhibited ≥90% inhibition; only four led to >70% cell viability (red rectangle in Figure 2c). These four compounds are pixantrone maleate, proflavine, quinacrine dihydrochloride, and QDD. A schematic representation of the screening procedure is shown in Figure 2d.

### 3.3. Dose-Dependent Inhibition of LASV MG Activity by Topoisomerase II Inhibitors

Four hit compounds were identified during our HTS. These compounds were pixantrone maleate, proflavine, quinacrine dihydrochloride and QDD. The criteria for selection were compounds which inhibited the LASV MG activity by at least 90% with a combined cytotoxicity of 30% or less. All four selected compounds inhibited LASV MG activity in a dose-dependent manner (Figure 3a). The IC_50_ values for pixantrone maleate, proflavine, quinacrine dihydrochloride, and QDD were 1.01 µM, 1.04 µM, 1.67 µM, and 1.28 µM, respectively. Pixantrone maleate, a known topoisomerase II inhibitor, had the highest selectivity index (SI) value of 9.21.

It is recommended that a prospective therapeutic agent should have an SI value ≥10 before being considered for further studies [23]. Due to the fact that pixantrone maleate was our hit compound which had the closest SI value to 10, we decided to select and assess the impact of other topoisomerase II inhibitors on the LASV MG system. The selected topoisomerase II inhibitorsinclude idarubicin hydrochloride, ellipticine hydrochloride, voreloxin (SNS 595) hydrochloride, and amonafide. They recorded IC_50_ values of 0.06 µM, 0.76 µM, 1.82 µM, and 2.89 µM, respectively (Figure 3b). Ellipticine hydrochloride exhibited an SI value of 13.51, making it the only compound with an SI >10 of all the compounds tested with the LASV MG.

### 3.4. Dose-Dependent Inhibition of JUNV MG Activity by Topoisomerase II Inhibitors

JUNV MG activity was initially validated with MPA and ribavirin, with IC_50_ values of 0.42 µM and 38.19 µM, respectively (Figure 4a). Our hit compounds also inhibited JUNV MG activity; pixantrone maleate, proflavine, quinacrine dihydrochloride, and QDD had IC_50_ values of 1.81 µM, 1.36 µM, 1.51 µM, and 1.46 µM, respectively (Figure 4b). These values, as well as the CC_50_ values recorded, were close to those observed with the LASV MG, with a maximum variation of ±3.0 µM. The only significant differences were that ribavirin had a 2.5 times higher IC_50_ and quinacrine dihydrochloride the highest SI value, at 8.19.

Idarubicin hydrochloride, ellipticine hydrochloride, voreloxin (SNS 595) hydrochloride, and amonafide also inhibited JUNV MG activity with IC_50_ values of 0.12 µM, 0.76 µM, 1.15 µM, and 2.90 µM, respectively (Figure 4c). These values were also close to those recorded with the LASV MG. Furthermore, ellipticine hydrochloride showed the highest SI, at 16.41.

The pan-anti-arenaviral capacity of the topoisomerase II inhibitors was also tested against an LCMV MG which was tagged with an RFP reporter and under the control of a murine-pol-I promoter. As shown in Figure 4d, all tested topoisomerase II inhibitors limited LCMV MG activity at a concentration of 5 µM. Ellipticine hydrochloride showed the highest impact of a 95% inhibition. These findings suggested that the topoisomerase II inhibitors are active against arenaviral MGs irrespective of the reporter or promoter used in MG construction.

### 3.5. Restriction of Live JUNV Replication by Topoisomerase II Inhibitors

The positive controls, MPA and ribavirin, inhibited live JUNV replication, with IC_50_ values of 0.97 µM and 45.36 µM, respectively (Figure 5a). Pixantrone maleate, proflavine, quinacrine dihydrochloride and QDD limited JUNV replication, with IC_50_ values of 3.66 µM, 2.45 µM, 7.33 µM, and 4.66 µM, respectively (Figure 5b), as did idarubicin hydrochloride, ellipticine hydrochloride, voreloxin (SNS 595) hydrochloride, and amonafide, with IC_50_ values of 2.56 µM, 1.73 µM, 5.23 µM, and 5.75 µM, respectively (Figure 5c). These results further indicate that the hit compounds in our MG-based screening and other topoisomerase II inhibitors are potential panarenaviral replication inhibitors.

### 3.6. Restriction of rLCMV.Arm by Topoisomerase II Inhibitors

Ribavirin, a positive control, restricted recombinant LCMV infection with an IC50 value of 20.19 µM (Figure 6a). Pixantrone maleate, idarubicin hydrochloride, ellipticine hydrochloride, voreloxin (SNS 595) hydrochloride, and amonafide did likewise with IC50 values of 4.75 µM, 0.59 µM, 1.50 µM, 8.5 µM and 10 µM, respectively (Figure 6b).

### 3.7. Limitation of Live JUNV Replication by siRNA Knockdown of Topoisomerase II

As shown in Figure 7a,b, expression of topoisomerases was effectively knocked down by the siRNAs. Based on band intensity, the siRNAs against topoisomerase IIβ had higher knockdown efficiencies. Figure 7c,d prove that the siRNAs were specific in their knockdown action. The topoisomerase IIα siRNA candidates knocked down only topoisomerase IIα as shown by the intracellular RNA levels (Figure 7c). This was similar to observations with the topoisomerase IIβ siRNA candidates as well (Figure 7d). This suggests that any co-dependence in expression between topoisomerases IIα and IIβ (as indicated by the immunoblot results) occurs during translation.

Knockdown of topoisomerase expression also had an impact on JUNV replication and propagation in culture supernatants with the topoisomerase IIα siRNAs TOPO2A siRNA 1 and TOPO2A siRNA 2 suppressing JUNV GPC RNA levels within the supernatant by 92% and 98.5%, respectively (Figure 7e) and the topoisomerase IIβ siRNAs TOPO2B siRNA 1 and TOPO2B siRNA 2 by 70% and 94%, respectively (Figure 7f). Conversely, the impact on intracellular JUNV GPC RNA levels was not as profound (Figure 7g,h); there was little or no reduction in comparison. We also observed that topoisomerase knockdown had minimal impacts on cell viability (Figure 7i,j). Putting these findings together, we postulate that the anti-arenaviral impact of topoisomerases II inhibition and/or knockdown occurs during the translation stage of the viral replication cycle. These results also suggest that JUNV replication requires expression of both topoisomerase IIα and IIβ, which may serve as molecular targets of host-oriented pan-arenavirus therapeutics.

### 3.8. Topoisomerase II Inhibitors Do Not Reduce Levels of the LASV MG Components

In order to prove that the topoisomerase II inhibitors only inhibit the reporter signals and do not affect the production of the LASV MG system components, we assessed the intracellular RNA levels of LASV NP, LASV L, and LASV MG after transfection and compound treatment. Huh-7 cells were seeded on a 24-well plate at a concentration of 4 × 10^5^ cells/mL and reverse transfected with the LASV MG system plasmids. At 12 h post-seeding, the cells were treated with selected topoisomerase II inhibitors at a concentration of 5 µM and ribavirin (100 µM). Intracellular RNA was collected 36 h post-treatment and used for qPCR assessment. Figure 8a–c show that there was no significant reduction in the RNA levels of LASV NP, LASV L, and LASV MG across the compound-treated wells. This suggests that the MG inhibitory effect observed is specific to the reporter signals (Figure 3).

Furthermore, the topoisomerase II inhibitors did not decrease intracellular RNA levels of the LASV MG (+) strand (Figure 8d). This suggests that transcription of the LASV MG RNA from the encoding plasmid was not hindered by compound treatment. The (-) strand intracellular RNA levels were also not reduced (Figure 8e), indicating that the compound treatment did not limit genome replication. Thus, the inhibitory activities of the compounds occur at post-transcription.

## 4. Discussion

Research on the development of specific countermeasures against arenaviruses and other pathogens causing viral hemorrhagic fever (VHF) has been limited over the years due to a strict requirement of high-containment facilities [24]. Systems that can mimic each viral replication step have helped to alleviate this bottleneck. One of them is the MG, a very powerful tool for modeling viral genome replication and transcription with potential for antiviral research outside high-containment facilities [15]. The arenaviral MG is usually complemented with plasmids expressing the arenaviral NP and L proteins. This is because the two proteins have been proven to be sufficient for effective genomic transcription and replication of arenaviruses [25].

In this study, we sought to identify novel inhibitors of LASV genome replication with potential anti-panarenaviral activity. First, we screened an FDA-compound library with a MG system derived from an LASV lineage II strain. Approximately 80% of confirmed cases during annual outbreaks are in LASV lineage II-endemic regions [26,27], which informed our decision to choose a strain from a relatively recent outbreak. Moreover, medical countermeasures developed based on presently circulating strains are more likely to be effective. Although the SI value of our lead hit compound pixantrone maleate is low in comparison to other HTS studies, its therapeutic potential should not be overlooked. In fact, derivatives of this compound maintain strong inhibitory ability with minimal cytotoxicity.

Topoisomerase II inhibitors comprise a group of compounds widely used as antineoplastic agents against many malignancies including cancers of the bladder and breast [28]. The observation that all topoisomerase II inhibitors restricted LCMV and JUNV replication suggests that some anticancer agents may also function as anti-arenaviral and the pathways by which they exert their antitumor activity may be crucial for arenaviral replication. Further studies with the infectious virus would be needed as our observations were with a recombinant virus and the Candid #1 vaccine strain, respectively. Previous studies have demonstrated that topoisomerase II inhibitors activate type-I interferon (IFN) signaling as the mechanism of antitumor and antiviral activities [29,30]. It was also reported that arenaviruses have the ability to evade host immune responses by inhibiting the production of type-I IFNs via viral NP-dependent degradation of immune stimulatory RNAs or through other mechanisms [31,32,33]. Nonetheless, topoisomerase II inhibitors have been proven to facilitate production of IFN in the presence of VP35, which is the IFN-antagonistic protein of EBOV [34]. Thus, we logically propose that topoisomerase II inhibitors likely exert anti-arenaviral effect through nullification of the immune evasion function of arenaviral NPs via potent stimulation of type-I IFN production. Another possible anti-arenaviral mechanism of action of the topoisomerase II inhibitors is the limitation of the translation stage of the arenavirus life cycle. This is supported by the facts that the LASV MG system components were still expressed in the presence of these compounds (Figure 8), while the reporter signals were efficiently knocked down (Figure 3).

Type II topoisomerases are part of a group of enzymes responsible for maintaining the topological state of DNA in a cell [35]. The enzymes exist in two discrete isoforms, topoisomerase IIα and IIβ, which are encoded by separate genes and have different physiological functions [36]. Nevertheless, our results suggest that expression of these two isoforms is not mutually exclusive, as knockdown of either topoisomerase silenced expression of the other (Figure 7a,b). It is likely that the expression of both proteins may be interrelated despite being encoded by different genes and having distinct functions. The presence of a mutually exclusive knockdown at the intracellular RNA level (Figure 7c,d) suggests that any potential co-dependence between the topoisomerase II isoforms takes place during protein expression. Moreover, knockdown of each protein resulted in a drastic reduction in JUNV Candid #1 RNA levels (Figure 7e,f) suggesting that type II topoisomerases might be novel host factors for arenaviruses. However, our results showed that there was no corresponding drastic reduction in intracellular JUNV RNA levels after knockdown (Figure 7g,h). This suggests that the anti-arenaviral impact of a knockdown of topoisomerases II is at the translation stage. This is in consonance with the results obtained with the topoisomerase II inhibitors. It is established that topoisomerases are important for replication of some DNA viruses, such as African swine fever virus [37,38], Epstein-barr virus [39], hepatitis B virus [40], and Kaposi’s sarcoma-associated herpesvirus [41], as their replication was limited by topoisomerase inhibitors. Our study demonstrated that topoisomerase II also regulates RNA virus infection by supporting replication/transcription and/or translation of the viral RNA genome.

The quest for alternatives to ribavirin has inspired studies on arenaviral entry inhibitors [42,43,44]. We understand that screening with MG systems can identify compounds that target only viral genome replication/transcription, thus excluding compounds targeting other stages of the entire viral life cycle. However, a combinational therapy of arenaviral genome replication inhibitors (for example, the topoisomerase II inhibitors discovered in this study) with arenaviral entry blockers might resolve this problem. This would be similar to highly active antiretroviral therapy (HAART), which is currently applied in the management of acquired immunodeficiency syndrome (AIDS). Further studies would be needed to ensure that such combination would be synergistic with minimal cytotoxicity. Such combination therapy would not only surmount the challenges associated with the use of ribavirin but also reduce the concentration of compounds required for clinical use, facilitate maximal therapy efficiency with minimal cytotoxicity, and limit the chance of developing drug resistance. Furthermore, the antiviral drugs targeting panarenavirus are expected to be effective against the emerging viral diseases by novel arenaviruses.

## Figures and Tables

**Figure 1 viruses-15-00105-f001:**
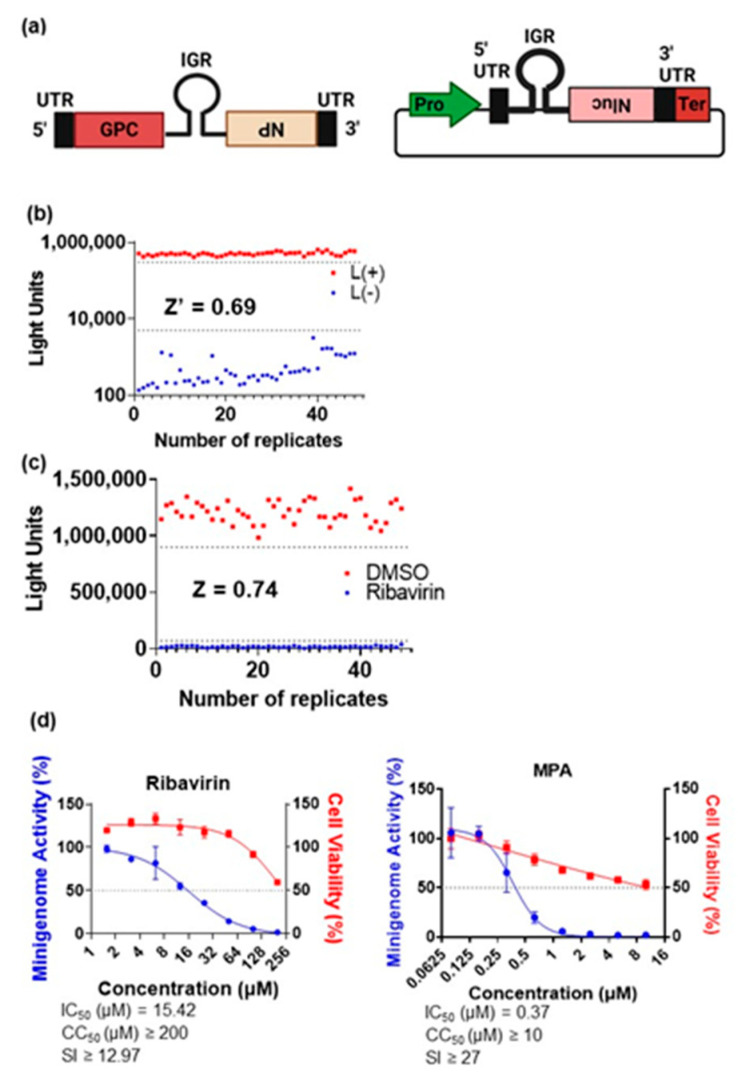
Construction, optimization and validation of the LASV MG. (**a**) Graphical representation of the original LASV S segment genome (left) and the LASV MG expressing the Nluc reporter, which was constructed based on the S segment (right). (**b**) Optimization of the quality of the constructed MG. Huh-7 cells were seeded on a 96-well plate and transfected with the LASV NP, LASV L, and LASV MG plasmids in order to assess the main signal of the system. The LASV L plasmid was replaced with an empty vector in order to measure the background signal. The MG signal was measured via Nluc expression at 48 h post-transfection. L(+) = MG assay with LASV polymerase, L(-) = MG assay without LASV polymerase. The Z’-factor was calculated as described in the Materials and Methods. (**c**) Optimization of the MG for high throughput screening (HTS). Huh-7 cells were seeded and transfected as described in (**b**). At 12 h post-transfection, half of the plate was treated with DMSO and the other half with 100 µM ribavirin. MG activity was measured at 36 h post-treatment. The Z-factor was calculated as described in the Materials and Methods. (**d**) Validation of the LASV MG. Huh-7 cells were seeded and transfected as described in (**b**). Two-fold serial dilutions of ribavirin and MPA in triplicate were added after 12 h. MG activity was assessed at 36 h post-treatment. The results were normalized to DMSO-treated wells. Pro—promoter; UTR—untranslated region; IGR—intergenic region; Nluc—nanoluciferase; Ter—terminator. Number of replicates on the x-axes of (**b**,**c**) refers to the number of wells dedicated to each of L(+), L(-), DMSO and ribavirin; these were 48 wells in each case.

**Figure 2 viruses-15-00105-f002:**
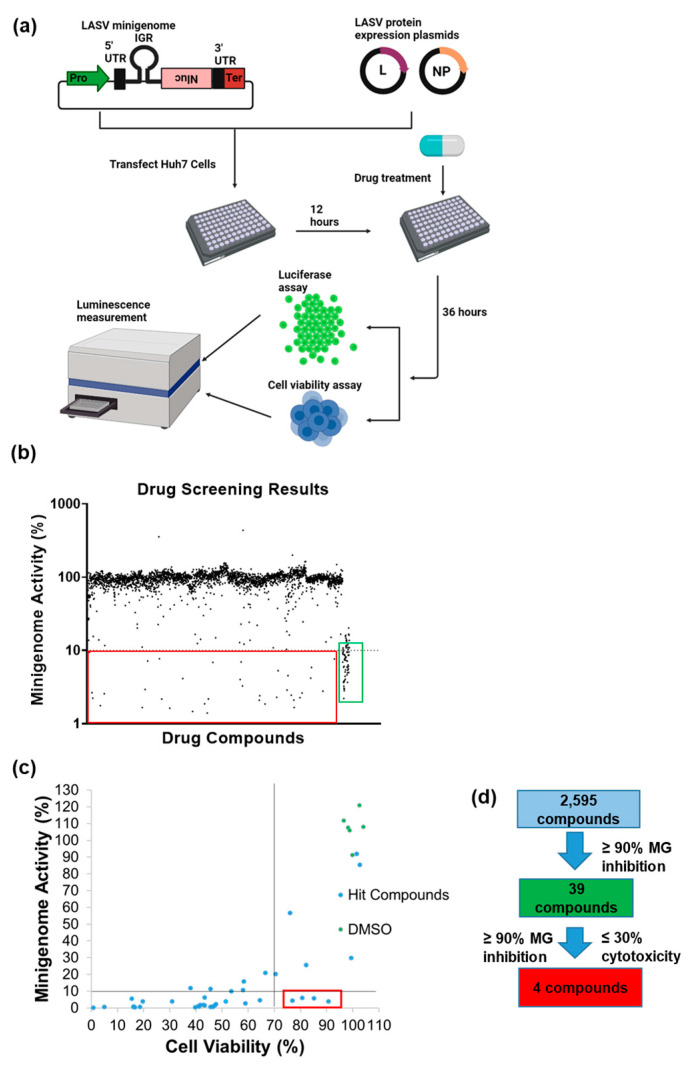
HTS for inhibitors of LASV MG activity using an FDA-approved drug library. (**a**) Schematic protocol of the compound screen. (**b**) HTS of a library of 2595 compounds for inhibitors of LASV MG activity. The percentile inhibition by each compound is represented by a black dot. Hit compounds (at a concentration of 5 µM) and positive controls (100 µM ribavirin and 5 µM MPA) are bordered by red and green rectangles, respectively. All assays were performed in duplicate simultaneously. The results were normalized to DMSO-treated wells. (**c**) Results of rescreening and cytotoxicity analyses of the hit compounds in (**b**). MG activity inhibition ≥90% and cell viability ≥70% were the criteria for the compounds selected for further analyses (bordered by a red rectangle). The results were normalized to DMSO-treated wells. (**d**) Schematic description of how the hit compounds were selected after the initial HTS and then for further analyses.

**Figure 3 viruses-15-00105-f003:**
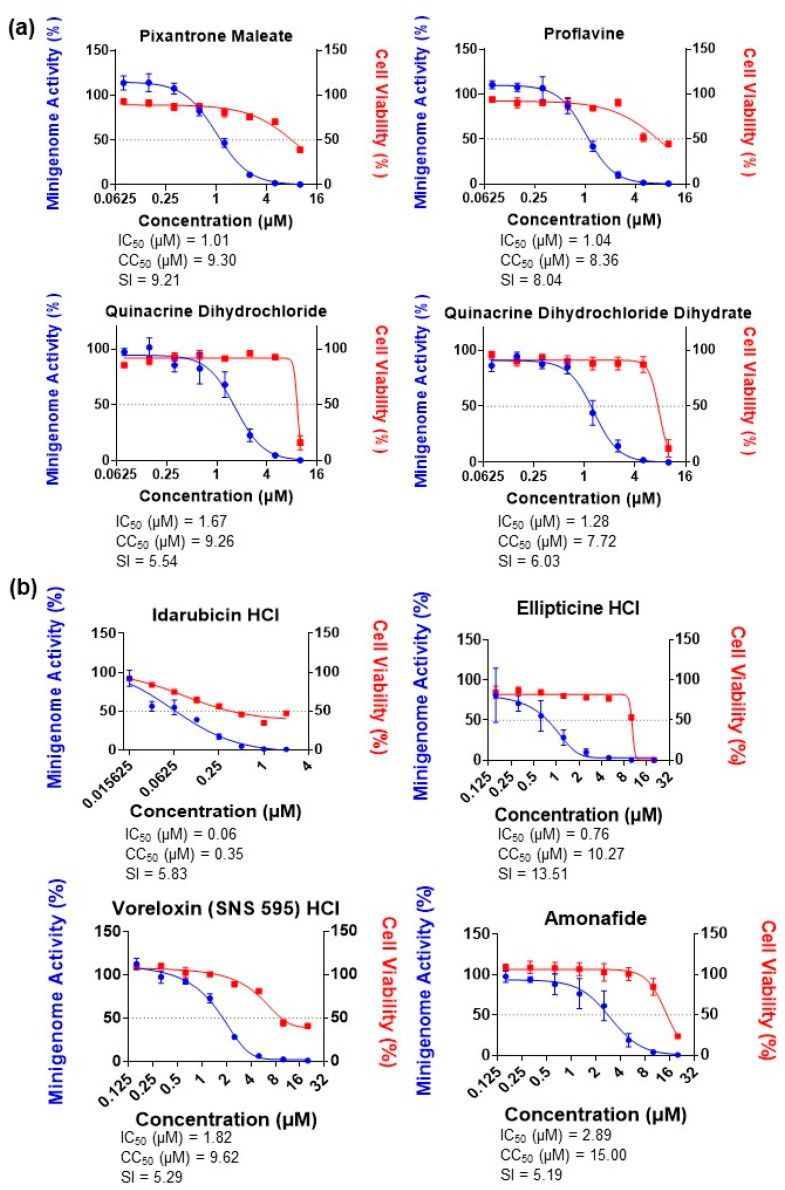
Inhibitory effects of identified candidates and topoisomerase II inhibitors against LASV MG. (**a**) Inhibitory effects of pixantrone maleate, proflavine, quinacrine dihydrochloride dihydrate, and quinacrine dihydrochloride against LASV MG. Huh-7 cells were transfected with the LASV MG, LASV NP, and LASV L plasmids. Compounds were added at the indicated concentrations in triplicate after 12 h. MG activity and cell viability were assessed at 36 h post-treatment. (**b**) Inhibitory effects of the topoisomerase II inhibitors idarubicin hydrochloride, ellipticine hydrochloride, voreloxin (SNS 595) hydrochloride, and amonafide against the LASV MG. Compound treatment and assessment were performed as described in (**a**). Data are presented as means ± SD and are representative of two independent experiments. All results were normalized to DMSO-treated wells.

**Figure 4 viruses-15-00105-f004:**
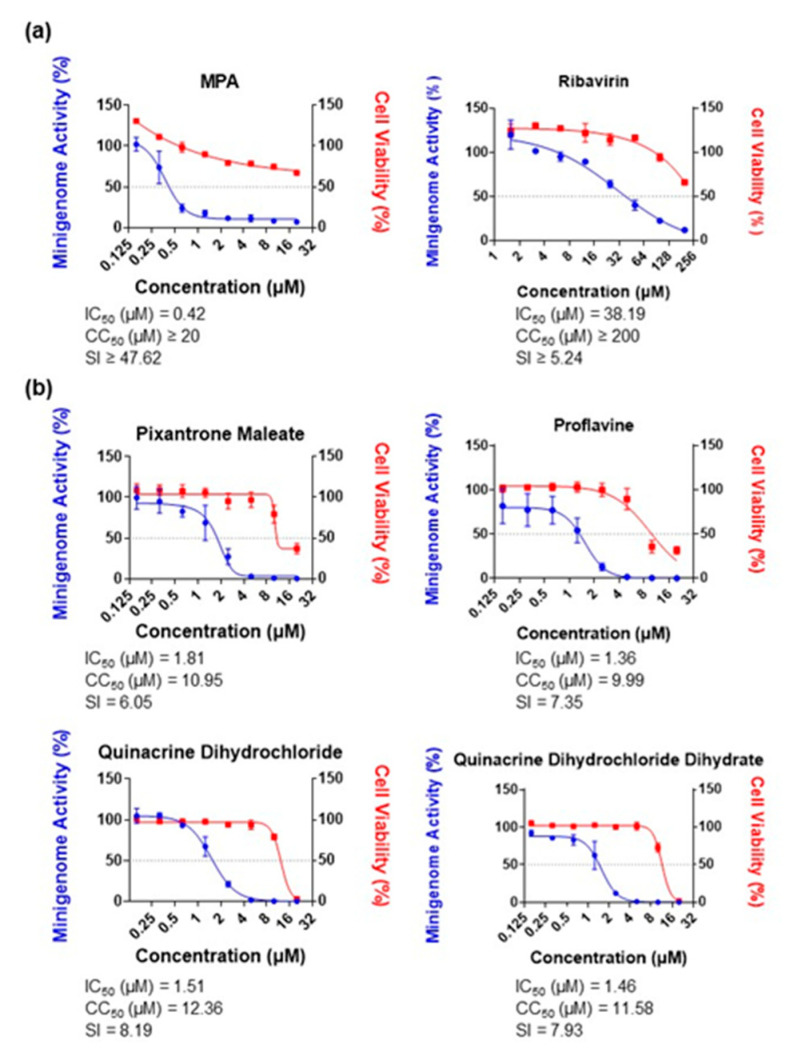
Inhibitory effects of identified candidates and topoisomerase II inhibitors against JUNV MG. (**a**) Inhibitory effects of the positive controls MPA and ribavirin against JUNV MG. Huh-7 cells were transfected with the JUNV MG, JUNV NP, and JUNV L plasmids. Compounds were added at the indicated concentrations in triplicate after 12 h. MG activity and cell viability were assessed at 36 h post-treatment. (**b**) Inhibitory effects of pixantrone maleate, proflavine, quinacrine dihydrochloride, and quinacrine dihydrochloride dihydrate against JUNV MG. Compound treatment and assessment were performed as described in (**a**). (**c**) Inhibitory effects of the topoisomerase II inhibitors idarubicin hydrochloride, ellipticine hydrochloride, voreloxin (SNS 595) hydrochloride, and amonafide against JUNV MG. Compound treatment and assessment were performed as described in (**a**). (**d**) Inhibitory effects of the topoisomerase II inhibitors pixantrone maleate, ellipticine hydrochloride, voreloxin (SNS 595) hydrochloride, and amonafide against LCMV MG. BHK-21 cells were transfected with the LCMV MG, LCMV NP, and LCMV L plasmids. Compounds were added at 5 µM in octuplicate after 12 h; Ribavirin (100 µM) was used as a positive control. MG activity was assessed at 36 h post-treatment. Data are presented as means ± SD and are representative of three independent experiments. All results were normalized to DMSO-treated wells.

**Figure 5 viruses-15-00105-f005:**
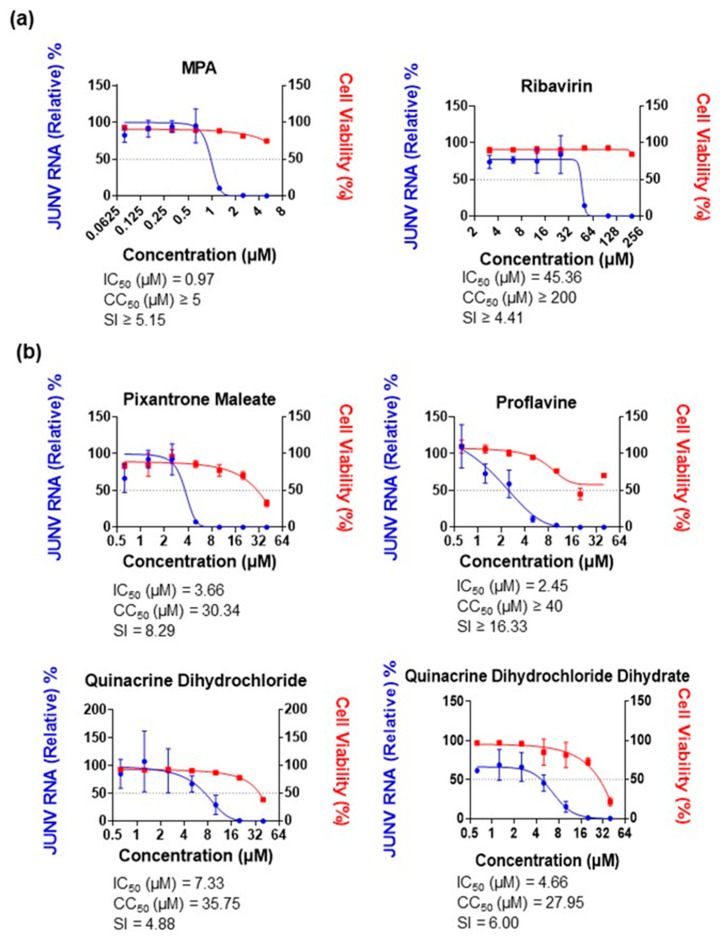
Inhibition of replication of JUNV by identified candidates and topoisomerase II inhibitors. (**a**) Restriction of JUNV replication by positive controls, MPA and ribavirin. A549 cells were seeded in 24-well plates and infected with JUNV at an MOI of 0.01 at 48 h post-seeding. The virus was washed out at 1 h.p.i, and the cells were incubated with compounds at the indicated concentrations in triplicate for 48 h. The supernatant was harvested and assessed by qPCR. Cell viability was also measured at the same time point. (**b**) Restriction of JUNV replication by the identified candidates pixantrone maleate, proflavine, quinacrine dihydrochloride, and quinacrine dihydrochloride dihydrate. JUNV infection, compound treatment, and assessment were performed as described in (**a**). (**c**) Restriction of JUNV replication by the topoisomerase II blockers idarubicin hydrochloride, ellipticine hydrochloride, voreloxin (SNS 595) hydrochloride, and amonafide. JUNV infection, compound treatment, and assessment were performed as described in (**a**). Data are presented as means ± SD and are representative of two independent experiments. Values in untreated wells were set to 100%.

**Figure 6 viruses-15-00105-f006:**
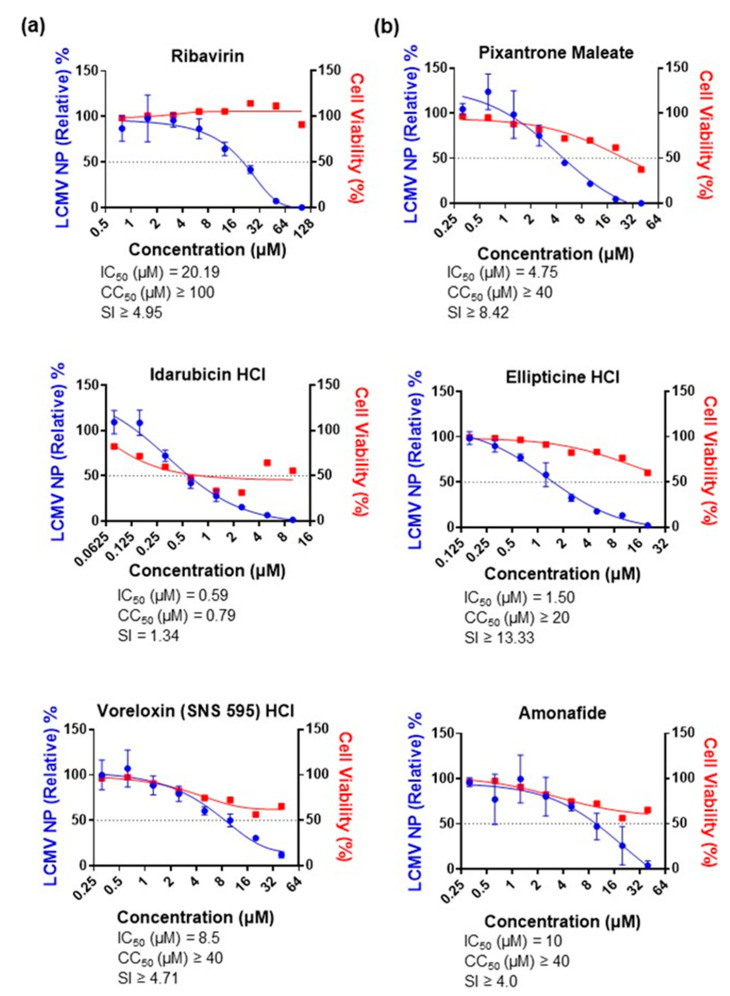
Inhibition of LCMV infection by ribavirin and topoisomerase II inhibitors. (**a**) Restriction of LCMV infection by ribavirin. A549 cells were seeded in 96-well plates and infected with LCMV (rLCMV.Arm) at an MOI of 0.1 at 24 h post-seeding. The virus was replaced with fresh medium at 1 h.p.i, and the cells were incubated with the compound at the indicated concentrations in triplicate for 24 h. The cells were then fixed, blocked, stained with anti-LCMV NP for 3 h, and later with secondary antibody for another 3 h before fluorescence was measured. (**b**) Restriction of LCMV infection by the topoisomerase II inhibitors pixantrone maleate, idarubicin hydrochloride, ellipticine hydrochloride, voreloxin (SNS 595) hydrochloride, and amonafide. LCMV infection, compound treatment, and assessment were performed as described in (**a**). Data are presented as means ± SD and are representative of two independent experiments. Values in untreated wells were set to 100%.

**Figure 7 viruses-15-00105-f007:**
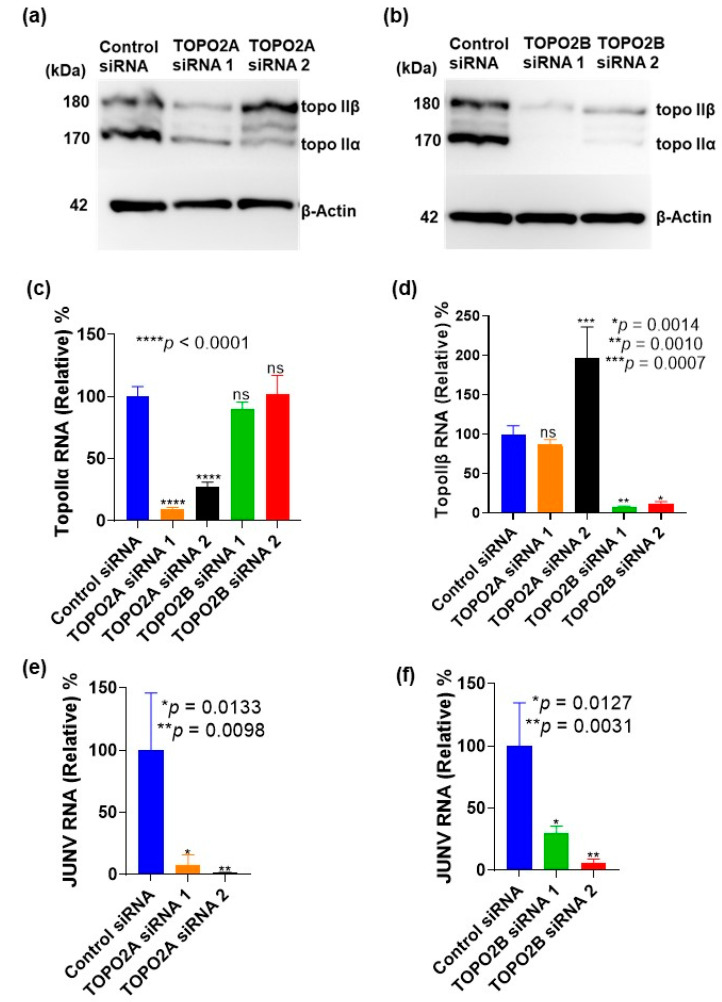
Knockdown of topoisomerase IIα and IIβ restricts JUNV replication. Representative immunoblot images of topoisomerase IIα (**a**) and IIβ (**b**) expression in A549 cells treated with control or topoisomerase IIα- or IIβ-targeting siRNAs. β-Actin was used as an internal control. A549 cells were seeded in 24-well plates and transfected with the indicated siRNAs simultaneously. The medium was replaced 3 days post-seeding, and the cell lysate was collected 2 days later. Intracellular RNA levels of topoisomerase IIα (**c**) and IIβ (**d**) after siRNA knockdown. A549 cells were seeded and transfected as described in (**a**,**b**). Intracellular RNA was collected for qPCR 3 days post-seeding. Impact of topoisomerase IIα siRNA (**e**) and IIβ siRNA (**f**) knockdown on JUNV replication. A549 cells were seeded and transfected with the indicated siRNAs in triplicate as described above. The cells were infected 3 days post-seeding, and the virus was washed out after 1 h of adsorption. The viral supernatants were collected at 2 days post-infection (d.p.i). JUNV GPC RNA levels in the supernatants were quantified with qPCR. The intracellular RNA was also collected 2 d.p.i to evaluate the impact of topoisomerase IIα siRNA (**g**) and IIβ siRNA (**h**) knockdown on intracellular RNA levels of JUNV GPC by qPCR. Effect of topoisomerase IIα siRNA (**i**) and IIβ siRNA (**j**) knockdown on A549 cell viability. Cells were seeded and transfected with the indicated siRNAs in triplicate as described above. Media was replaced on the 4th day of the experiment and cell viability was measured at 2 days after the media was replaced. Data are presented as means ± SD and are representative of two independent experiments. P values are provided in the graphs. Values in control siRNA wells were set to 100%.

**Figure 8 viruses-15-00105-f008:**
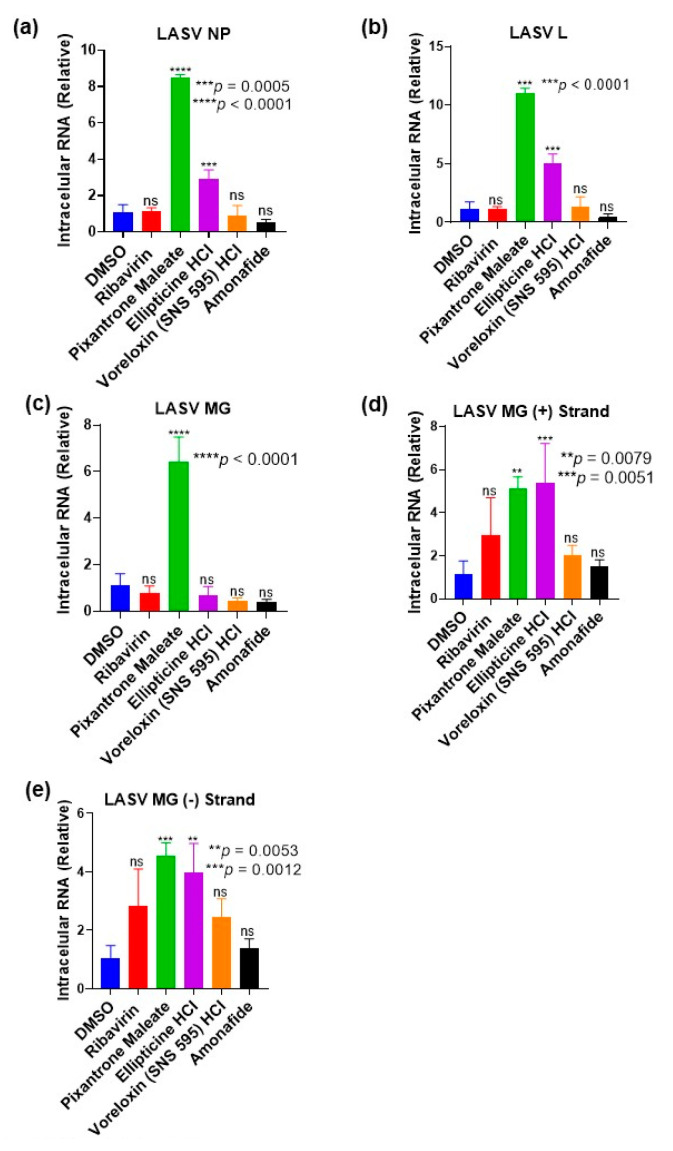
Topoisomerase II inhibitors do not reduce the expression of LASV MG system components. Huh-7 cells were seeded and reverse-transfected with LASV MG system plasmids on a 24-well plate. At 12 h post-seeding, topoisomerase II inhibitors (5 µM) and ribavirin (100 µM) were introduced in triplicate. Intracellular RNA was collected at 36 h post-treatment for qPCR analyses of (**a**) LASV NP, (**b**) LASV L, and (**c**) LASV MG via one-step RT-qPCR. Intracellular RNA levels of (**d**) LASV MG (+) strand and (**e**) LASV MG (-) strand were determined using two-step RT-qPCR. Data are presented as means ± SD. Values are relative to expression in DMSO-treated wells.

## Data Availability

All data that support the figures and findings of this study are available upon reasonable request from the corresponding author.

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
