# Peer review of "Topoisomerase II as a Novel Antiviral Target against Panarenaviral Diseases"

_viruses, 2022, doi:10.3390/v15010105_

Round 1

Reviewer 1 Report

In this manuscript, Authors found pixantrone maleate, a known topoisomerase II inhibitor, as a hit in screening, a good candidate against arenaviral diseases. Author used RNA polymerase I-driven minigenome (MG) expression system for Lassa virus (LASV) for high-throughput screening (HTS). This manuscript identified multiple topoisomerase II inhibitors, also inhibited Junín virus (JUNV) MG activity and siRNA knockdown of both topoisomerases (IIα and IIβ) restricted JUNV replication. This is a well-written manuscript, author did a great effort, but I suggest some changes that must be made to the text, contribute to a better understanding of the points they are trying to make. Overall, the manuscript is improved but there are still some points the authors should consider.

Major comments

1.     In Figure 1a, Author shows graphical representation of constructed LASV MG. Please show the graphical representation of original genomic structure of Lassa virus, so that the construct will be more understandable for the readers.

2.     Legends of Figure 1 are little bit confusing please explain in a better way.

3.     In Figure 1, minigenome activity was measured after 36 hours post transfection whereas in figure 3 and 4, minigenome activity was measured after 48 hours post transfection. Can author please explain why you did 36 hours treatment in figure 1.

4.     Author can also show the role of topoisomerase in arenaviral replication by illustration, so that it will be more convincing and understandable for the readers. Manuscript

5.     Figure 3 showed the impact of topoisomerase II inhibitors, including idarubicin hydrochloride, ellipticine hydrochloride, voreloxin (SNS 595) hydrochloride, and amonafide. Ellipticine hydrochloride exhibited an SI value of 13.51, making it the only compound with an SI >10. Author should explain little bit more about four positive hits in screening and topoisomerase II inhibitors.

1.     In Figure 6, viral supernatant was collected for RNA extraction and qPCR after Knockdown of topoisomerase IIα and IIβ to see the effect on JUNV (Junín virus) replication. Author can also check the level of intracellular RNA by cell lysis. Figure 6a and 6b showed knockdown of topoisomerase IIα or IIβ restricts JUNV replication. The expression of topoisomerase showed by western blot. Author can also show by qPCR, so that it will be more convincing.

2.     In line 316 and 317, author explained about limitation of siRNA knockdown. Yes, immunoblot is not convincing for topoisomerase IIα. Actin band of negative siRNA have higher intensity as compared to TOPO2A. Author can also use the siRNA pool instead of single siRNA for better knockdown.

3.     In Figure 6c, JUNV RNA level is not visible when TOPO2A siRNA is used. Author can use line bar horizontally for better representation.

Author Response

For Reviewer 1

In this manuscript, Authors found pixantrone maleate, a known topoisomerase II inhibitor, as a hit in screening, a good candidate against arenaviral diseases. Author used RNA polymerase I-driven minigenome (MG) expression system for Lassa virus (LASV) for high-throughput screening (HTS). This manuscript identified multiple topoisomerase II inhibitors, also inhibited Junín virus (JUNV) MG activity and siRNA knockdown of both topoisomerases (IIα and IIβ) restricted JUNV replication. This is a well-written manuscript, author did a great effort, but I suggest some changes that must be made to the text, contribute to a better understanding of the points they are trying to make. Overall, the manuscript is improved but there are still some points the authors should consider.

Re: Thank you very much for your comments regarding our manuscript. We revised our manuscript according to your comments. The followings are our reply to the points that you mentioned.

Major comments

  1. In Figure 1a, Author shows graphical representation of constructed LASV MG. Please show the graphical representation of original genomic structure of Lassa virus, so that the construct will be more understandable for the readers.

Re: According to your suggestion, Figure 1a has been modified.

  1. Legends of Figure 1 are little bit confusing please explain in a better way.

Re: This has been adjusted and we hope it is more comprehensible now (Legends of Figure 1).

  1. In Figure 1, minigenome activity was measured after 36 hours post transfection whereas in figure 3 and 4, minigenome activity was measured after 48 hours post transfection. Can author please explain why you did 36 hours treatment in figure 1.

Re: We apologize for the typographical errors. Legends of Figures 3 and 4 were modified accordingly. For clarity, minigenome activity was always measured at 36 hours post-treatment. The only situation where 48 hours post-transfection arises was when the main and background signals were measured. These have been appropriately reflected on the manuscript (Legends of Figures 3 and 4).

  1. Author can also show the role of topoisomerase in arenaviral replication by illustration, so that it will be more convincing and understandable for the readers. Manuscript

Re: We appreciate your suggestion. Although topoisomerase II was found to regulate intracellular replication steps of arenaviruses, we have not identified its detailed role. Therefore, in order to refrain from misleading readers, we decided not to show the illustration. Hypothesized roles of topoisomerase II were described in discussion section in the manuscript. We hope this is understandable.

  1. Figure 3 showed the impact of topoisomerase II inhibitors, including idarubicin hydrochloride, ellipticine hydrochloride, voreloxin (SNS 595) hydrochloride, and amonafide. Ellipticine hydrochloride exhibited an SI value of 13.51, making it the only compound with an SI >10. Author should explain little bit more about four positive hits in screening and topoisomerase II inhibitors.

Re: We appreciate your important comment and can fully understand the point. Although we have already explained the suggested point in the original manuscript, we have included the additional discussion about the selection of the four positive hits and how topoisomerase II inhibitors were selected for in-depth study in the revised manuscript (lines 294-310 on page 9).

  1. In Figure 6, viral supernatant was collected for RNA extraction and qPCR after Knockdown of topoisomerase IIα and IIβ to see the effect on JUNV (Junín virus) replication. Author can also check the level of intracellular RNA by cell lysis. Figure 6a and 6b showed knockdown of topoisomerase IIα or IIβ restricts JUNV replication. The expression of topoisomerase showed by western blot. Author can also show by qPCR, so that it will be more convincing.

Re: According to your suggestions, we performed a series of experiments.

At first, we found that the intracellular RNA levels of JUNV GPC showed no or slight reduction by both TOPO2A and TOPO2B siRNAs (Figures 7g and 7h, lines 420-422 on page 16, the “Results” section 3.7.). It will be recalled that all siRNAs elicited a significant reduction in JUNV GPC RNA levels within the supernatant. A combined interpretation of these results suggests that the impact of topoisomerase II knockdown on JUNV replication and propagation is mainly at the translation stage.

Second, the expression of topoisomerase observed by western blot was further investigated by qPCR as suggested. The results showed that the siRNAs were specific in their knockdown function at the intracellular RNA level; no cross-knockdown was observed as seen in the western blot results (Figures 7c and 7d, lines 410-415 on the page 16 and 17, the “Results” section 3.7.). This suggests that the cross-reaction observed only takes place during the protein expression. The reason(s) for this is not known.

  1. In line 316 and 317, author explained about limitation of siRNA knockdown. Yes, immunoblot is not convincing for topoisomerase IIα. Actin band of negative siRNA have higher intensity as compared to TOPO2A. Author can also use the siRNA pool instead of single siRNA for better knockdown.

Re: We opine that difference in actin bands is a reflection of the cell viability which is shown graphically in Figure 7i. However, the reduction of the cell viability (actin band intensity) is much smaller than that of infectivity, indicating that the effects of siRNA knockdown is still specific.

Our intracellular RNA results showed that both TOPO2A siRNA 1 and TOPO2A siRNA 2 were quite efficient in knocking down topoisomerase IIα. The results of pooling TOPO2A siRNA 1 and siRNA 2 was not as strong as using TOPO2A siRNA 1 alone (data not shown). Thus, we decided to maintain the status quo of two siRNA candidates for each topoisomerase isoform. We also believe that having two candidates each with the same impact consolidates our submissions.

  1. In Figure 6c, JUNV RNA level is not visible when TOPO2A siRNA is used. Author can use line bar horizontally for better representation.

Re: We appreciate your suggestion. However, this does not make a difference as this JUNV RNA level is close to zero.

Reviewer 2 Report

Oladipo Afowowe et al. show that by using an arenaviral minigenome (MG) system, they could successfully identify topoisomerase II inhibitors which affect viral entry. This paper follows previous research which demonstrated a connection between viral propagation and host topoisomerases. These findings open the door to obtaining a new panarenaviral drug that may outperform ribavirin and MPA. This manuscript should be published, but with additional data (especially with the proper negative controls for the MG assay) and further clarifications in the text.

Major comments:                                             

·       A major concern with this MG activity assay is the lack of some controls. It remains unclear whether the inhibitory effect indeed reduce the replication of the MG, or alternatively affects the production of L or NP. Also, it is not clear if similar results will be obtained using a different reporter.

Proper controls could validate whether the MG is truly the source of the topoisomerase II-driven inhibition. First, the authors can demonstrate inhibition using a different reporter at the MG. Second, the authors should demonstrate that the inhibitors do not reduce the levels of L and NP (via RT-qPCR or WB).  

·       If indeed topoisomerase II is the true target of the inhibitors, then overexpressing it should suppress the effect of the drugs. Demonstrating that will strongly substantiate the claims of the authors. This could also be used to differentiate the role of the different topoisomerase isoforms.

·       “Based on band intensity, the siRNAs against topoisomerase IIβ had higher knockdown efficiencies.” It also seems that TOPO2B siRNA’s was more effective at topoisomerase IIα knockdown then was TOPO2A siRNA’s. Please mention why you think this is the case (besides these genes having an overlapping function)?

·       “However, a combinational therapy of entry blockers with genome replication inhibitors will resolve this problem”
The authors recommend the use of a combinatory treatment, in which a topoisomerase II inhibitor could be taken with ribavirin. Ribavirin is a chemotherapeutic-like drug which has serious side effects. The authors should at least acknowledge that such a combinatory therapy may be too toxic and that the possibility for its future use as a therapy is speculative.

If you choose to keep this recommendation in the discussion section, then I suggest providing additional data to help substantiate your claim (such as providing CC50 data for the combinatory treatment), but nonetheless, please clarify that administering this combinatory therapy is speculative and could be highly toxic to humans.

·       “Restriction of Live JUNV Replication by Topoisomerase II Inhibitors”.
The Candid #1 strain of the Junin virus is a live attenuated virus – but with significantly diminished virulence to the human host. “The observation that all topoisomerase II inhibitors restricted JUNV replication suggests that some…”
The authors should be cautious with assuming that the live virus acts in the same manner as Candid #1. If they wish to make this claim, then additional experiments with live JUNV viruses should be carried out. Otherwise, please emphasize in the manuscript that the JUNV results pertain to Candid #1. Such data may also be representative of the live virus, but additional work would be needed to verify this claim.

Minor Comments:

·       Please mention why it was chosen to co-express the NP with the L protein.

·       I recommend clarifying what “Number of Replicates” refers to, on the x-axis is Figure 1.

·       Please include in the text that other topoisomerase II drugs have also been shown to inhibit viral infection (PMID: 24821256, 27568922, 22106228, 30867306)

·       “Moreover, knockdown of each protein resulted in a drastic reduction in JUNV Candid #1 RNA levels (Figure 6c and 6d) suggesting that type II topoisomerases might be novel important host factors for arenaviruses”.
Please remove either “novel” or “important”.

·       Please mention if there were any replicates for the HTS of the drug candidates.

·       Please discuss in the text if you observed any significant differences on the effect (via IC50 & CC50) of the studied drugs on the New World (JUNV) arenaviral polymerase compared to the Old World (LASV) polymerase? 

Author Response

For Reviewer 2

Oladipo Afowowe et al. show that by using an arenaviral minigenome (MG) system, they could successfully identify topoisomerase II inhibitors which affect viral entry. This paper follows previous research which demonstrated a connection between viral propagation and host topoisomerases. These findings open the door to obtaining a new panarenaviral drug that may outperform ribavirin and MPA. This manuscript should be published, but with additional data (especially with the proper negative controls for the MG assay) and further clarifications in the text.

Re: Thank you very much for your comments regarding our manuscript. We revised our manuscript according to your comments. The followings are our reply to the points that you mentioned.

Major comments:                                             

  • A major concern with this MG activity assay is the lack of some controls. It remains unclear whether the inhibitory effect indeed reduce the replication of the MG, or alternatively affects the production of L or NP. Also, it is not clear if similar results will be obtained using a different reporter.

Proper controls could validate whether the MG is truly the source of the topoisomerase II-driven inhibition. First, the authors can demonstrate inhibition using a different reporter at the MG. Second, the authors should demonstrate that the inhibitors do not reduce the levels of L and NP (via RT-qPCR or WB).  

Re: According to your suggestions, a series of experiments were performed. At first, in order to prove that the inhibition observed is not reporter-dependent, we tested the topoisomerase II inhibitors with the LCMV minigenome tagged with a mCherry (RFP) reporter and driven by a murine pol-I promoter. Our observation that all compounds also significantly inhibited the LCMV minigenome expression suggests that minigenome inhibition is consistent irrespective of the reporter or promoter used (Figure 4d, lines 337-343 on the pages 10-11, the “Results” section 3.4.).

Second, we found that intracellular RNA levels of LASV NP and LASV L after compound treatment remained unchanged. This finding suggests that the MG is the source of the inhibition observed (Figure 8, lines 450-460 on the page 19, the “Results” section 3.8.).

  • If indeed topoisomerase II is the true target of the inhibitors, then overexpressing it should suppress the effect of the drugs. Demonstrating that will strongly substantiate the claims of the authors. This could also be used to differentiate the role of the different topoisomerase isoforms.

Re: We appreciate your helpful comments. The target of these compounds have been established in literature as topoisomerase II. It should also be noted that these compounds have different structures but still exert the same impact, suggesting that topoisomerase II is the molecular target of the inhibitors in the context of arenavirus replication.

In addition, while we did not overexpress the protein, we suppressed it by using siRNAs. This was confirmed with our western blot experiments. The fact that the suppression of the protein also restricted JUNV replication and propagation consolidates our claim, in our opinion. 

  • “Based on band intensity, the siRNAs against topoisomerase IIβ had higher knockdown efficiencies.”It also seems that TOPO2B siRNA’s was more effective at topoisomerase IIα knockdown then was TOPO2A siRNA’s. Please mention why you think this is the case (besides these genes having an overlapping function)?

Re: It is difficult to arrive at an assertive reason without more specific experiments. However, from literature, we know that topoisomerase IIβ makes up 90% of the topoisomerases in cancer cells. If our hypothesis that the expression of the proteins is inter-dependent is true, a knockdown on topoisomerase IIβ will likely have a higher impact. Furthermore, the intracellular RNA levels of the topoisomerases after knockdown showed that the siRNAs were specific in action (Figures 7c and 7d). This also suggests that the cross-knockdown observations were at the point of protein expression. Once again, an inter-dependency in protein expression is possible.

  • “However, a combinational therapy of entry blockers with genome replication inhibitors will resolve this problem”
    The authors recommend the use of a combinatory treatment, in which a topoisomerase II inhibitor could be taken with ribavirin. Ribavirin is a chemotherapeutic-like drug which has serious side effects. The authors should at least acknowledge that such a combinatory therapy may be too toxic and that the possibility for its future use as a therapy is speculative. 

If you choose to keep this recommendation in the discussion section, then I suggest providing additional data to help substantiate your claim (such as providing CC50 data for the combinatory treatment), but nonetheless, please clarify that administering this combinatory therapy is speculative and could be highly toxic to humans.

Re: The combinational therapy we actually proposed here is that of topoisomerase II inhibitors with entry blockers, but not ribavirin. As you indicated, we do admit that this suggestion is speculative and have adjusted the paragraph to reflect this (Page 22, lines 536-538, the fifth paragraph in the “Discussion” section).

  • “Restriction of Live JUNV Replication by Topoisomerase II Inhibitors”.
    The Candid #1 strain of the Junin virus is a live attenuated virus – but with significantly diminished virulence to the human host. “The observation that all topoisomerase II inhibitors restricted JUNV replication suggests that some…”
    The authors should be cautious with assuming that the live virus acts in the same manner as Candid #1. If they wish to make this claim, then additional experiments with live JUNV viruses should be carried out. Otherwise, please emphasize in the manuscript that the JUNV results pertain to Candid #1. Such data may also be representative of the live virus, but additional work would be needed to verify this claim.

Re: These observations are well acknowledged and have been reflected in the manuscript (lines 493-494 on the page 21, the third paragraph in the “Discussion” section).

Minor Comments:

  • Please mention why it was chosen to co-express the NP with the L protein.

Re: These two proteins have been proven to be sufficient for effective genomic transcription and replication of arenaviruses. This piece of information has also been inserted into the text in the “Discussion” section (Page 21, lines 478-481, the first paragraph in the “Discussion” section).

  • I recommend clarifying what “Number of Replicates” refers to, on the x-axis is Figure 1.

Re: This refers to the number of wells of the 96-well plate dedicated to each experimental group. This has been further clarified in the legend of Figure 1.

  • Please include in the text that other topoisomerase II drugs have also been shown to inhibit viral infection (PMID: 24821256, 27568922, 22106228, 30867306)

Re: A couple of these references were already mentioned in the text to show the importance of topoisomerases to the replication of some mentioned DNA viruses. However, the narrative that the replication was restricted by topoisomerase inhibitors, as well as the other references, has been added (Pages 23-24, lines 529-533 in the “Discussion” section).

  • “Moreover, knockdown of each protein resulted in a drastic reduction in JUNV Candid #1 RNA levels (Figure 6c and 6d) suggesting that type II topoisomerases might be novel important host factors for arenaviruses”.
    Please remove either “novel” or “important”.

Re: According to your suggestion, “important” has been deleted (Page 23, line 525).

  • Please mention if there were any replicates for the HTS of the drug candidates.

Re: The compound screening (HTS) was performed in duplicate. This was mentioned in the “Materials and Methods” section (Page 4, lines 144-146).

  • Please discuss in the text if you observed any significant differences on the effect (via IC50 & CC50) of the studied drugs on the New World (JUNV) arenaviral polymerase compared to the Old World (LASV) polymerase? 

Re: As we mentioned in the “Result” section 3.4. (Page 10, lines 324-331), the IC50 values of the studied topoisomerase II inhibitors on both the LASV and JUNV minigenome systems were quite close. We have also included a statement to cover the CC50 values as well (line 328). There was a maximum variation of ±3 µM. We did not observe any significant differences.

Reviewer 3 Report

In this manuscript, the authors began their studies by performing a drug inhibitor screen using a minigenome (MG) expression system of Lassa fever virus (LASV), a highly pathogenic arenavirus. Upon secondary validation of initial hits, they identified one compound known to inhibit the activity of topoisomerase II to further investigate. Similar topoisomerase inhibitors also inhibited the LASV MG system, as well as the Junin virus (vaccine strain) MG system. Additional studies with live Junin infections using siRNA and topoisomerase II inhibitor treatments were shown in support of the final conclusions of the study that assign a role in a DNA binding protein in RNA arenavirus infection. Overall, the study was clearly described and well written.

However, one of the major issues with the study is there is insufficient evidence to support the conclusions of the paper. Specific examples include:

1)    Given that topoisomerase II is DNA binding protein and an unconventional target for RNA virus infection, the MG system that is based on DNA plasmid transfection should have been further investigated to rule out the possibility of assay artifact. The cells dosed with specific inhibitors of viral replication in an MG based system should all still express the MG RNA and the arenaviral proteins, but only production of the reporter signal should be inhibited. Protein and RNA measurements should have been included in at least one of the drug inhibitor treatments to verify the inhibitors are not acting on the expression of the MG components.

2)    The drug selectivity index was a factor in down-selection of candidate inhibitors, however, all the studies show very low SI values, with mostly around or below 10 in the MG and viral infection assays. As the system gets more complex, MG expression vs in vitro inhibition vs in vivo inhibition, the SI values are likely to be reduced and resolution of any discrepancies between inhibition and toxicity would be very minimal.

3)    Only two arenavirus MG systems were tested with these inhibitors. These minimal amounts of systems tested cannot claim a panarenaviral disease inhibitor. 

4)    Lastly, the proposed mechanism of action described in the discussion involves modulating the innate immune defense which would not be specific to arenaviruses and therefore again the title doesn’t match the final conclusions. 

5)    Overall, the topoisomerase II inhibitors seem highly cytotoxic compared to ribavirin, a drug used in patients, which results in overall low SI values for drug development.

Minor comment:

1)    The section 3.4 title is mislabeled should read JUNV MG not LASV MG.

Author Response

For Reviewer 3

In this manuscript, the authors began their studies by performing a drug inhibitor screen using a minigenome (MG) expression system of Lassa fever virus (LASV), a highly pathogenic arenavirus. Upon secondary validation of initial hits, they identified one compound known to inhibit the activity of topoisomerase II to further investigate. Similar topoisomerase inhibitors also inhibited the LASV MG system, as well as the Junin virus (vaccine strain) MG system. Additional studies with live Junin infections using siRNA and topoisomerase II inhibitor treatments were shown in support of the final conclusions of the study that assign a role in a DNA binding protein in RNA arenavirus infection. Overall, the study was clearly described and well written.

However, one of the major issues with the study is there is insufficient evidence to support the conclusions of the paper. Specific examples include:

Re: Thank you very much for your comments regarding our manuscript. We revised our manuscript according to your comments. The followings are our reply to the points that you mentioned.

1) Given that topoisomerase II is DNA binding protein and an unconventional target for RNA virus infection, the MG system that is based on DNA plasmid transfection should have been further investigated to rule out the possibility of assay artifact. The cells dosed with specific inhibitors of viral replication in an MG based system should all still express the MG RNA and the arenaviral proteins, but only production of the reporter signal should be inhibited. Protein and RNA measurements should have been included in at least one of the drug inhibitor treatments to verify the inhibitors are not acting on the expression of the MG components.

Re: According to your suggestion, we performed additional experiments to demonstrate that intracellular RNA levels of LASV NP and LASV L as well as the MG RNA after compound treatment remained unchanged (Figure 8, lines 450-460 on the page 19, the “Results” section 3.8.). This finding suggests that only the reporter signal was inhibited in the treated wells.

2) The drug selectivity index was a factor in down-selection of candidate inhibitors, however, all the studies show very low SI values, with mostly around or below 10 in the MG and viral infection assays. As the system gets more complex, MG expression vs in vitro inhibition vs in vivo inhibition, the SI values are likely to be reduced and resolution of any discrepancies between inhibition and toxicity would be very minimal.

Re: We appreciate to your comment. This seems related to 5) and would be answered together.

3) Only two arenavirus MG systems were tested with these inhibitors. These minimal amounts of systems tested cannot claim a panarenaviral disease inhibitor. 

Re: We appreciate your helpful comment. We have included results which show that the topoisomerase II inhibitors also restrict LCMV MG expression as well as the replication of a recombinant LCMV (Figures 4d and 6, lines 337-343 and 386-390 on pages 10-11 and 14, respectively). This consolidates our claim and provides a good basis for further testing against other infectious arenaviruses.

4) Lastly, the proposed mechanism of action described in the discussion involves modulating the innate immune defense which would not be specific to arenaviruses and therefore again the title doesn’t match the final conclusions. 

Re: We appreciate your important suggestion. We opine that it is not unusual for a particular compound or group of compounds to elicit antiviral effects against more than one group of viruses. Our research was solely focused on arenaviruses. The compounds studied exhibited antiviral effects against all arenaviral platforms used. In addition, a knockdown of the suspected host factor, topoisomerase II in this case, restricted the replication of JUNV. Our conclusion is that therapy involving the use of compounds targeting this host factor should be explored. Thus, we believe that our title and conclusion are valid.

5) Overall, the topoisomerase II inhibitors seem highly cytotoxic compared to ribavirin, a drug used in patients, which results in overall low SI values for drug development.

Re: We understand that the topoisomerase II inhibitors are cytotoxic. However, we hypothesize a couple of solutions to this problem. One is to develop less cytotoxic derivatives of these compounds. We also propose research on a combinational therapy with arenaviral entry inhibitors, which might be able to decrease the doses of topoisomerase II inhibitors with minimizing the cytotoxicity and keeping the antiviral activity (lines 536-542 on page 22 in the “Discussion” section). In conclusion, despite the high cytotoxicity observed with these compounds, we believe that we cannot discard the fact that the study has revealed a novel host factor for efficient arenaviral replication.

Minor comment:

 1)    The section 3.4 title is mislabeled should read JUNV MG not LASV MG.

Re: According to your suggestion, we corrected it (line 323 at the page 10). Thank you.

Reviewer 4 Report

The manuscript “Topoisomerase II as a Novel antiviral target against panarenaviral disease” by Oladipo Afowowe et al.,  describes Topoisomerase II as a suitable target against arenavirus infections. Inhibitor studies as well as siRNA knockdown experiments of Toposiomerases IIα and IIβ reduced viral replication of JUNV. These findings open the door for Pixantrone maleate to be used as novel drug candidate against arenavirus infections. The manuscript it well written and the data is conclusive.

Major comments:

1.       I do not understand why the authors choose to analyze the MG activity in Fig. 3,4,5 after 48h but in Fig.1 after 36h. Wouldn´t it be more consistent to choose 48 h for all the MG activity experiments?

2.       Why did the authors do “at least” two independent experiments in Fig. 3-6? Why not 3 independent experiments? In the most ideal way, the experiments are done three times but the authors should at least mention the exact number of independent experiments.

Minor comments:

1.       I would not write that “a combinational therapy of entry blockers with genome replication inhibitors will resolve the problem.” It might resolve the problem. One can not definitely say that two drugs will work together unless a synergy test was done. They might act antagonistic on each other.

Author Response

For Reviewer 4

The manuscript “Topoisomerase II as a Novel antiviral target against panarenaviral disease” by Oladipo Afowowe et al., describes Topoisomerase II as a suitable target against arenavirus infections. Inhibitor studies as well as siRNA knockdown experiments of Toposiomerases IIα and IIβ reduced viral replication of JUNV. These findings open the door for Pixantrone maleate to be used as novel drug candidate against arenavirus infections. The manuscript it well written and the data is conclusive.

Re: Thank you very much for your comments regarding our manuscript. We revised our manuscript according to your comments. The followings are our reply to the points that you mentioned.

Major comments:

  1. I do not understand why the authors choose to analyze the MG activity in Fig. 3,4,5 after 48h but in Fig.1 after 36h. Wouldn´t it be more consistent to choose 48 h for all the MG activity experiments?

Re: It was actually a typographical error and has been corrected in the text (Legends of Figures 3a and 4a).

  1. Why did the authors do “at least” two independent experiments in Fig. 3-6? Why not 3 independent experiments? In the most ideal way, the experiments are done three times but the authors should at least mention the exact number of independent experiments.

Re: We appreciate to your helpful suggestion. We performed only two independent experiments when we got quite similar results from them. If not, we always performed three independent experiments to ensure the reproducibility of the results. According to your suggestion, the exact number of independent experiments have been shown for each study (Legends of Figures 3-7).

Minor comments:

  1. I would not write that “a combinational therapy of entry blockers with genome replication inhibitors will resolve the problem.” It mightresolve the problem. One can not definitely say that two drugs will work together unless a synergy test was done. They might act antagonistic on each other.

Re: Your comment is quite appropriate. We reflected it on the text (Page 22, lines 532-534, in the “Discussion” section).

Round 2

Reviewer 3 Report

Dear Authors,

Thank you for addressing some of the major concerns from the original version of the manuscript. This version includes some important data to strengthen your conclusions, specifically with the addition of inhibitor treatment against recombinant LCMV (Fig 6) and also measurements of MG component expression through RT-qPCR (Fig 8). 

One last question remains to improve clarity of the methods described for Fig 8. As I understand the MG assay, DNA plasmids are transfected into cell to express the NP,  L and the nLuc-MG reporter genes separately. Expression of the negative RNA strand of the nLuc-MG cassette is dependent on the polI system, while the expression of the positive RNA strand of the nLuc reporter is dependent on the activity of the NP and L genes involved in viral replication. In figure 8, it's important to distinguish the relative levels of RNA of the (-)RNA LASV MG from the (+) nLuc mRNA as the topo II inhibitors should only reduce the (+) nLuc mRNA levels and not the (-)RNA LASV MG. From the methods (2.11) describing a one-step RT-qPCR method, this would not be able to distinguish (+) from (-) strand RNA of the MG. 

As a majority of the paper is based on MG data, it's important to have these controls highly accurate. A properly designed two-step RT-qPCR method might be beneficial to perform prior to publication.

Interestingly, the topo II inhibitor Pixantrone Maleate significantly increased expression of all the MG components (Fig 8), but did not reduce them as the authors pointed out.

Outside of this one recommendation, the manuscript has been significantly improved. 

Author Response

For Reviewer 3

One last question remains to improve clarity of the methods described for Fig 8. As I understand the MG assay, DNA plasmids are transfected into cell to express the NP,  L and the nLuc-MG reporter genes separately. Expression of the negative RNA strand of the nLuc-MG cassette is dependent on the polI system, while the expression of the positive RNA strand of the nLuc reporter is dependent on the activity of the NP and L genes involved in viral replication. In figure 8, it's important to distinguish the relative levels of RNA of the (-)RNA LASV MG from the (+) nLuc mRNA as the topo II inhibitors should only reduce the (+) nLuc mRNA levels and not the (-)RNA LASV MG. From the methods (2.11) describing a one-step RT-qPCR method, this would not be able to distinguish (+) from (-) strand RNA of the MG. 

As a majority of the paper is based on MG data, it's important to have these controls highly accurate. A properly designed two-step RT-qPCR method might be beneficial to perform prior to publication.

Interestingly, the topo II inhibitor Pixantrone Maleate significantly increased expression of all the MG components (Fig 8), but did not reduce them as the authors pointed out.

Re: Thank you very much for your comments regarding our manuscript.

According to your suggestion, we performed a two-step RT-qPCR in order to assess the impact of topoisomerase II inhibitors on LASV MG (+) and (-) strands. The methodology for this can be found in lines 233-241 on page 5. The results and figures for this experiment on lines 471 to 476. The legend for Figure 8 has also been edited to reflect the new experiments. This can be seen in lines 483 to 485 on page 22. These are the major changes made to the manuscript and we hope that you find them satisfactory.

Reviewer 4 Report

I would like to thank the authors for replying to my suggestions. I think that the manuscript can be published in its present form.

Author Response

For Reviewer 4

I would like to thank the authors for replying to my suggestions. I think that the manuscript can be published in its present form.

Re: Thank you very much for reviewing our manuscript. We appreciate your comments and the time you took to review our manuscript.